# Relationship between socio-demographics, body composition, emotional state, and social support on metabolic syndrome risk among adults in rural Mongolia

Jin Hee Kim[1], Hyun Lye Kim[1], Bolorchimeg Battushig[2], Jae Yong Yoo[1]*

1 Department of Nursing, College of Medicine, Chosun University, Gwangju, South Korea, 2 Department of Nursing, Medical School, Mongolian National University, Ulaanbaatar, Mongolia

* jaeyongyoo@chosun.ac.kr

## Abstract

### Background

In Mongolia, where there is a large regional gap in the quality of healthcare services, metabolic syndrome (MetS) is steadily increasing. However, there are few studies on the risk level of MetS and affecting factors among adults living in rural Mongolia. This study aims to explore the relationship between socio-demographics, clinical characteristics, emotional state, and social support on the risk level of MetS prevalence among adults living in rural Mongolia.

### Methods

In this cross-sectional study, 143 adults living in the soum area of Dondgovi aimag in Mongolia were recruited. Data collection was conducted from July 2 to 3, 2019. The self-reported questionnaires including socio-demographic, clinical characteristics and emotional status, anthropometric tests using the InBody, and blood sampling tests were conducted. The number of individual diagnostic criteria met was scored as a MetS risk score and classified into 6 groups, from the lowest score of 0 to the highest score of 5. The ordinal logistic regression analysis was used to identify the factors affecting the risk of MetS.

### Results

The prevalence of MetS among adults living in rural Mongolia was 58.0%, and the mean MetS risk score was 2.70±1.34 points. In the ordinal logistic regression analysis, age, regular exercise of moderate intensity or higher, InBody score reflecting obesity or sarcopenia, and depression level were statistically significantly associated with the risk score for MetS.

### Conclusions

Our study demonstrated that MetS risk levels among adults living in rural Mongolia with limited medical resources were strongly associated with demographic characteristics, body composition and emotional health condition, particularly depression.

**Data Availability Statement:** All relevant data are within the paper and its Supporting Information files.

**Funding:** This research was supported by Leading University Project for International Cooperation through the National Research Foundation of Korea (NRF) funded by the Ministry of Education (MOE) (NO. NRF-2016H1A7A2A02913910). The funders had no role in study design, data collection and analysis, decision to publish, or preparation of the manuscript.

**Competing interests:** The authors have declared that no competing interests exist.

## Introduction

Metabolic syndrome (MetS) is a chronic metabolic disorder of cardiovascular risk factors, such as abdominal obesity, hyperglycemia, dyslipidemia, and hypertension, which are complex expressions [1]. MetS has a high prevalence rate worldwide not only due to aging, but also due to lifestyle practices, such as a lack of physical activity, smoking, excessive drinking, and overnutrition [2]. In the United States [3], it has been reported that 3 out of 5 adults—and 54.9% of those over age 60—have MetS. According to epidemiological studies in Asia, 24.2% of adults in China [4], 30.5% in Korea [5] and 32.8% of those in Mongolia [6] have MetS. In addition, the prevalence rate of MetS tends to rise as the elderly population increases and exposure to risk factors continues [5, 7, 8]. Adults with MetS are more than twice as likely to develop cerebrovascular and cardiac diseases, and have various complications such as diabetes and dyslipidemia [9]. Therefore, health professionals have consistently emphasized the importance of the systematic prevention and management of MetS. In previous studies [4, 6, 10–13], the prevalence of MetS was reported to be higher in adults and postmenopausal women with a high body mass index (BMI), high blood pressure, dyslipidemia, type 2 diabetes, and cardiovascular disease. It has also been reported that the risk of MetS increases when one's socioeconomic level and educational level are low [8, 14], when one smokes [15], frequently binge drinks [16], and does less physical activity [17, 18]. Recent studies [19–23] have emphasized the importance of understanding both psychosocial factors and physical characteristics. In particular, it was found that emotional states such as stress and depression, health-related lifestyle habits such as regular exercise, and environmental factors such as social support also affect the prevalence of MetS.

MetS has a variety of characteristics that depend on race, region, gender, and age [24, 25]. For systematic MetS management, it is important to understand not only one's physical characteristics, but also one's demographic characteristics, clinical characteristics (e.g., body composition), and social and psychological situations. Most of the previous studies relating to MetS were conducted in Western countries, such as the United States [7, 26] and Europe [27], or were focused on Korea [5, 28], Japan [29, 30], and China [4, 18, 31] among the Asian countries. Within Asia, Mongolia has a unique nomadic lifestyle, with a meat-oriented diet and frequent drinking culture [32]. In addition, the prevalence of non-communicated diseases, including MetS, has rapidly increased therein due to an environment where physical activity is restricted—due to a continental climate characterized by cold, long winters and short summers [16]. In Mongolia, nearly 67% of the population lives in the capital, Ulaanbaatar, while around 33% others still live in traditional houses (gers/yurts) in rural areas [33]. Non-communicated diseases accounted for about 80% of all deaths in Mongolia from 2006 to 2016, and the prevalence of the five major diseases increased by 1.3 to 2.2 times over the same time span [34]. Cardio-cerebrovascular disease accounts for 40% of all deaths in Mongolia, which is reported to be the 7th highest among countries in the Western Pacific region [34]. The "Health in All Policies" approach to the Sustainable Development Goals [35] emphasizes that all people should be equally provided with quality healthcare services. Despite such efforts in Mongolia, the lack of strategies for the prevention and management of non-communicated diseases, the lack of medical service support for vulnerable groups (e.g., elderly, women, children), and the gap in access to health care services between the capital and rural areas are challenges that have yet to be resolved [36, 37]. Indeed, many people in the country are migrating from rural to urban areas due to the accessibility of health care services [38]. This is becoming more common as people in the rural areas continue to age, and the weakening of the social support base becomes more serious [37]. Therefore, the basis for establishing strategies for the prevention and management of MetS based on an understanding of the regional, social, and environmental characteristics of rural communities in Mongolia are required to address these challenges.

Therefore, this study investigated the risk of MetS in adults residing in rural areas of Mongolia and explored the relationship between their socio-demographics, clinical characteristics (e.g., body composition), emotional state, and social support level on the risk of MetS prevalence.

## Materials and methods

### Study design and participants

This descriptive cross-sectional study explored the relationship between socio-demographics, clinical characteristics (e.g. body composition), emotional state, and social support on the risk level of MetS prevalence among adults living in rural Mongolia. The subjects of this study were adults over 20 years old living in the soum area of Dondgovi aimag (province), about 260 km south of Ulaanbaatar, the capital of Mongolia. Subjects that were mostly limited in their daily lives due to severe diseases, were suffering from degenerative diseases such as terminal cancer, or were unable to communicate due to cognitive impairment were excluded from this study. Convenience sampling was carried out in consideration of the purpose of the research. This study was conducted in collaboration with researchers at the Mongolian National University as a part of the International Cooperation Leading University Promotion Support Project under the Ministry of Education of Korea and National Research Foundation of Korea. In order to recruit rural residents, we received cooperation from government officials in the soum district office. Two weeks before the collection of data, an explanatory text written in Mongolian was distributed to each household through a letter-based notice. One week before data collection, a second notice was provided to encourage voluntary participation and proper compliance (i.e., to fast for at least 8 hours before visiting the test site for accuracy of blood tests). The data collection period was a total of 2 days from July 2 to 3, 2019. The total number of adult residents in the study area was about 450, and 188 visited the study site. Among them, a total of 143 subjects were analysed—excluding 5 subjects who were under the age of 20, 6 subjects who refused to or did not perform an anthropometric measurement, and 34 subjects who did not undergo a blood test or had a test error.

### Measurements

All questionnaires were translated and back-translated into Mongolian. The research team consulted a Mongolian linguist and two nursing professors, all of whom were bilingual. One Korean lives in Mongolia and works as a professor of Korean studies at the Mongolian National University, and two Mongolian nursing professors have lived in Korean for years. They determined whether there was a difference in meaning. And if there is any disagreement in the translation-reverse translation process, the agreement was reached through discussion. The Mongolian version of the questionnaire has been further revised and supplemented with comments from experts consisting of two nursing professors and one social welfare professor. According to the results of a pilot study of 10 Mongolian residents, the Mongolian questionnaire was finally completed after being revised into easy-to-understand terms.

**Socio-demographics and health-related behavior characteristics.** The socio-demographics included age, gender, occupation, education level, monthly income, and marital status. For characteristics relating to health-related behavior, the drinking, smoking, physical activity, and sleep pattern items of the Korean National Health and Nutrition Examination Survey (KNHANES) were utilized [39]. Drinking items included drinking frequency and average amount of alcohol at a time; and smoking items included actual smoking status, average daily smoking amount, and smoking cessation plan. Using the International Physical Activity Questionnaire, physical activity was classified as "yes" when high-intensity exercise or

moderate exercise was regularly performed. Otherwise, it was classified as "no." Sleep pattern was surveyed using the question, "How many hours do you usually sleep in a day?"

**Clinical characteristics, body composition, and blood test.** *Anthropometric and clinical characteristics.* Height and weight were measured with a height-weight scale (RGZ-200, China) in an upright position with shoes off. According to the World Health Organization (WHO) classification [40], a body mass index of $\geq 25$ kg/m$^2$ is classified as obesity. Waist and hip circumference measurements were measured by a well-trained researcher using a tape measure. The subject's top was raised and measured after maintaining normal breathing and upright posture. Systolic and diastolic blood pressure were measured with an automatic sphygmomanometer (HEM7121, Omron, Japan) after resting for 10 minutes in a sitting position. Under clinical characteristics, the items on the presence of comorbidity were classified as "yes" if the subject was diagnosed with cerebrovascular disease, cardiac disease, pulmonary disease, hypertension, diabetes mellitus, and chronic liver disease. There was also a question asking whether the subject was currently taking medication related to those diseases.

Subjects removed their shoes and socks, wore light everyday clothes, and measured their body composition using the InBody 270 (Biospace, Korea), which performs bioelectrical impedance analysis. Body composition measurements included total body water, protein, minerals, body fat mass, free fat mass, skeletal muscle mass, obesity index, InBody score, and basal metabolic rate. Subjects were instructed to maintain a fasting state for more than 8 hours before the test, and not to engage in vigorous physical activity or alcohol. Prior to the body composition measurement, the subject's bladder was emptied, and all metals and socks that affect current flow were removed.

For the accuracy of the blood test, blood was collected after fasting for at least 8 hours. Serum total cholesterol and triglyceride levels were measured using a Cholestech LDX (Alere Inc., Waltham, MA, USA) analyser, and fasting blood glucose was measured using an Accu-check Go (Roche Diagnostics, Mannheim, Germany). Cholestech LDX analysis, an automatic analysis system, was performed by two medical personnel trained in strict standardized operating procedures. The subject's hand was wiped with an alcohol swab, dried, and then blood was collected and analysed through a 40μL capillary using a lancet. An optic check was performed daily for quality control of the Cholestech LDX analyser.

**Metabolic syndrome risk scores.** The risk score for MetS was evaluated in accordance with the diagnostic criteria of the National Cholesterol Education Program Adult Treatment Panel III (NCEP-ATP III) [1] and the classification guidelines for serum total cholesterol results [1]. The criteria for classification of risk level for MetS are as follows: 1) waist circumference: male $\geq 90$ cm, female $\geq 85$ cm (based on the Asia-Pacific region), 2) hypertension: systolic/diastolic blood pressure $\geq 130/85$ mmHg; 3) high blood sugar: fasting blood sugar $\geq 100$ mg/dL; 4) hypertriglyceridemia: blood triglyceride $\geq 150$ mg/dL; and 5) high total cholesterol $\geq 200$ mg/dL. It was determined that the subjects currently taking medications related to each indicator had an abnormality in metabolic risk. Due to administrative restrictions, such as the prohibition of bringing test reagents into Mongolia, this study used the index of total cholesterol in blood instead of HDL cholesterol, which is mainly included as a diagnostic criterion for MetS. If more than three of the five diagnostic criteria were present, the subject was classified as having MetS. In addition, the number of individual diagnostic criteria met was scored as a MetS risk score and classified into 6 groups, from the lowest score of 0 to the highest score of 5.

**Emotional state.** Emotional state included the subject's subjective health status, usual stress perception, and depression level.

For subjective health status, the question was: "How do you usually think of your health?" Answers were on a 5-point scale, ranging from 1 (very good) to 5 (very poor).

For stress perception, the question was: "How much stress do you usually feel?" Answers could be ranked as follows: 4 points for 'I feel very much,' 3 points for 'I feel a lot,' 2 points for 'I feel a little,' and 1 point for 'I rarely feel any.' The subjects who responded with 'I feel a lot' or 'I feel very much' were classified into the group that perceived stress.

The Patient Health Questionnaire-9 (PHQ-9) [41] was used to determine whether the subjects had depressive symptoms. The PHQ-9 is a tool developed by Spitzer in a self-report format for the diagnosis of mental disorders that can be easily seen in primary care institutions; it is mainly a tool for evaluating depressive symptoms. It consists of a total of 9 questions, and was designed according to the diagnostic criteria for depressive episodes of the Diagnostic and Statistical Manual of Mental Disorders, Fourth Edition (DSM-IV). Each item is calculated as 0–3 points, for a total of 27 points. 0–4 points indicates non-depression, 5–9 points indicates mild depression, 10–19 points indicates depression, and 20–27 points indicates severe depression.

**Social support.** The level of social support was measured using a social support tool developed by Park [42]. In the pilot test with five Mongolian residents, six questions were excluded that were difficult for locals to understand or that advisory professors suggested to delete because there were duplicate expressions. A tool consisting of the remaining 19 questions was used. The social support tool consists of four subdomains (emotional support, informational support, material support, and appraisal support), and each item consists of a 5-point Likert scale ranging from 1 point representing 'not at all' to 5 points representing 'very much.' After summing the scores of each question, the average was obtained by dividing the sum by the number of questions in the subdomain. The higher the score, the higher the social support. Construct validity was confirmed, and the reliability coefficient Cronbach's α was .94 at the time of development of the tool [42], and .82 in this study.

## Data collection and ethical considerations

Data collection was conducted from July 2 to 3, 2019 in Dondgovi aimag, Mongolia. The self-reported questionnaires, anthropometric tests, and blood sampling tests were conducted in the main auditorium at the Office of Education in Dondgovi aimag. The questionnaire survey was conducted face-to-face by three Mongolian researchers who were trained to conduct the survey in the same manner as three nursing professors. When the subject was illiterate or elderly and had difficulty reading the questionnaire, the researchers were allowed to read each sentence to be answered. The anthropometric measurements, blood collection, and analysis were performed by medical staff with doctor or nursing licenses. The purpose, method, and importance of the study were explained to the subjects who wished to participate in the study, and their written consent was obtained. The total time required was 25 minutes—comprising 10 minutes for the questionnaire, 10 minutes for the anthropometric and body composition measurements, and 15 minutes for blood tests. In order to reduce the boredom of the subjects waiting for the next process to be performed, a health education video for the prevention and management of MetS that was translated into Mongolian was provided for viewing.

The study was conducted in accordance with the Declaration of Helsinki, and the study protocol was approved by the C University Hospital Institutional Review Board (IRB number: 2018-10-003). Informed written consent was obtained from all of the study participants. Physical measurements and blood collection were performed by trained medical staff, and local medical staff from Mongolia were present to prevent unexpected accidents. Rest areas and chairs were provided in order to prevent accidents due to movement limitations, fatigue, or falls of elderly subjects. In addition, local researchers who could speak Mongolian were deployed and guided the participants one-on-one. No accidents occurred during the study

period. Individual health counseling was provided by Mongolian medical staff regarding the results after examination, and adults with high MetS risk scores were advised to undergo additional health checkups.

## Statistical analysis

This study used the SPSS 25.0 program for data analysis, and the analysis method was as follows. First, a descriptive statistical analysis was conducted to determine the level of variables related to the sociodemographic and clinical characteristics, body composition, blood test, emotional state, and social support of the subjects. Chi-test and ANOVA analyses were performed to confirm the differences according to the MetS risk score. Next, ordinal logistic regression analysis was used to identify the factors affecting the risk of MetS. Ordinal logistic regression analysis is similar to multinomial logistic regression analysis in that it analyses the probability of selection of a dependent variable with three or more items. However, when testing a case where the dependent variable is of the sequence type, ordinal logistic regression analysis is considered to be more suitable [43].

The steps to interpret ordinal logistic regression are as follows: first, an ordinal logistic regression analysis was performed through the parallel lines assumption. Thereafter, a test of the null hypothesis that the regression coefficient values were the same for each category of the dependent variable was performed. After evaluating the fit of the entire model through the likelihood ratio test, the variables that were found to have a significant influence on the dependent variable among the independent variables were identified if the fit of the model was statistically significant. The odds ratio (OR) was also calculated. All statistical significance levels were set at $p < .05$.

## Results

### General characteristics of subjects

The mean age of the 143 subjects was 49.3±13.2 years, and 84.6% of the subjects were female. Among the five diagnostic criteria for MetS, 55.2% of the subjects had abdominal obesity, 65.0% had hypertension, 38.5% had high blood sugar, 63.6% had high triglyceride, and 30.1% had high total cholesterol. It was found that 58.0% of all subjects met three or more of the diagnostic criteria for MetS. The mean MetS risk score was 2.70±1.34 points (Table 1).

Due to the rural setting, most of the subjects (93.7%) had nomadic and agricultural jobs. 88.8% of them lived with their spouses. As for the characteristics of health-related behaviors, 58.0% of the subjects usually drink alcohol, 7.7% currently smoke, and 16.1% and 31.5% of them regularly engage in high-intensity and moderate exercise, respectively. There was an average of 0.99 comorbid diseases, of which hypertension was diagnosed to be the most common (33.6%), followed by chronic liver disease (21.7%) and cardiac disease (20.3%). Among the medications currently being taken, antihypertensive medications were the most common (28.7%). The average systolic blood pressure was around 128.0mmHg and the diastolic blood pressure was approximately 86mmHg. According to the BMI classification criteria [40], 72.8% of the subjects were above 25.0kg/m², and 31.5% of them were above 30.0kg/m².

The body composition analysis showed that total body water was 30.74kg, protein 8.21kg, minerals 2.85kg, body fat mass 27.24kg, free fat mass 41.81kg, and skeletal muscle mass was 22.82kg, respectively. Among them, 91.6% of the subjects had excess body fat mass. More than 91% of all subjects had excess body fat. In the analysis of the abdominal obesity level, 76.9% were found to be obese, with a level of 10 or higher. Upon summarizing the InBody score, 69.2% were found to be obese or lacking in muscle, and the average basal metabolic rate was 1273.18±164.29 kcal.

Among the emotional states, 91.6% of subjects answered that their subjective health status was moderate to very poor, and 77.0% were experiencing stress in their daily life. According to

**Table 1. Descriptive statistics of study subjects (*N* = 143).**

| Variables | n (%) /M±SD | Variables | n (%) /M±SD |
|---|---|---|---|
| **Socio-demographics** | | Waist circumference (cm) | 88.46±11.75 |
| Gender | | Waist-hip ratio | 0.95±0.07 |
| Women | 121 (84.6) | Systolic blood pressure (mmHg) | 128.01±17.23 |
| Men | 22 (15.4) | Diastolic blood pressure (mmHg) | 86.34 ±12.79 |
| Age (Years) | 49.3±13.2 | Fasting blood glucose (mg/dL) | 103.32±37.30 |
| 20–29 | 11 (7.7) | Total cholesterol (mg/dL) | 175.93±51.78 |
| 30–39 | 26 (18.2) | Triglyceride (mg/dL) | 231.90±154.98 |
| 40–49 | 31 (21.7) | Comorbidities | 0.99±1.02 |
| 50–59 | 38 (26.6) | Cerebrovascular disease | 5 (3.5) |
| 60–69 | 30 (21.0) | Cardiac disease | 29 (20.3) |
| ≥ 70 | 7 (4.9) | Pulmonary disease | 21 (14.7) |
| Educational level | | Hypertension | 48 (33.6) |
| Middle school or less | 29 (20.2) | Diabetes mellitus | 7 (4.9) |
| High school | 52 (36.4) | Chronic liver disease | 31 (21.7) |
| Diploma/Bachelors or above | 62 (43.4) | Current medications | 0.78±0.96 |
| Occupation | | Cerebrovascular disease | 4 (2.8) |
| Yes | 134 (93.7) | Cardiac disease | 24 (16.8) |
| No | 9 (6.3) | Pulmonary disease | 17 (11.9) |
| Monthly incomes (MNT) | 596,294.5±428,000.5 | Hypertension | 41 (28.7) |
| < 100,000 | 14 (9.8) | Diabetes mellitus | 5 (3.5) |
| 100,000 to 299,999 | 12 (8.4) | Chronic liver disease | 21 (14.7) |
| 300,000 to 499,999 | 39 (27.3) | | |
| 500,000 to 999,999 | 53 (37.1) | **Body compositions** | |
| ≥≥ 1,000,000 | 25 (17.5) | Total body water (kg) | 30.74±5.59 |
| Marriage status | | Shortage/ Excess | 15(10.5)/27(18.9) |
| Living with the spouse | 127 (88.8) | Protein (kg) | 8.21±1.52 |
| Living without the spouse | 16 (11.2) | Shortage/ Excess | 19(13.3)/25(17.5) |
| Current drinker | 83 (58.0) | Minerals (kg) | 2.85±0.50 |
| Current smoker | 11 (7.7) | Shortage/ Excess | 13(9.1)/28(19.6) |
| Regular exercise | | Body fat mass (kg) | 27.24±9.65 |
| High intensity exercise (yes) | 23 (16.1) | Shortage/ Excess | 0(0.0)/ 131(91.6) |
| Times per week | 2.7±1.2 | Free fat mass (kg) | 41.81±7.60 |
| Duration (minutes) | 55.8±19.2 | Shortage/ Excess | 15(10.5)/ 26(18.2) |
| Moderate intensity exercise (yes) | 45 (31.5) | Skeletal muscle mass (kg) | 22.82±4.58 |
| Times per week | 3.6±1.8 | Shortage/ Excess | 19(13.3)/ 30(21.0) |
| Duration (minutes) | 51.7±22.4 | Visceral fat level | |
| Sleeping time (hours) | 7.41±1.14 | < 10 levels (normal) | 33 (23.1) |
| **Clinical characteristics** | | ≥ 10 levels (obesity) | 110 (76.9) |
| Height | 156.71±7.31 | InBody score | |
| Weight | 69.06±14.58 | ≥ 70 (normal) | 44 (30.8) |
| Body mass index (kg/m$^2$) | 28.00±4.95 | < 70 (overweight/lack muscle) | 99 (69.2) |
| <18.5 (underweight) | 1 (0.7) | Body metabolic rate (kcal) | 1273.18±164.29 |
| 18.5–24.9 (normal) | 38 (26.5) | | |
| 25.0–29.9 (overweight) | 59 (41.3) | | |
| ≥ 30.0 (obese) | 45 (31.5) | | |
| **Metabolic syndrome status** | | Self-rate health status | 3.02±0.49 |
| Metabolic syndrome components | | Excellent to good | 12 (8.4) |

*(Continued)*

**Table 1.** (Continued)

| Variables | n (%) /M±SD | Variables | n (%) /M±SD |
|---|---|---|---|
| Abdominal central obesity (90/85cm) | 79 (55.2) | Average | 114 (79.7) |
| High blood pressure (130/85mmHg) | 93 (65.0) | Not so good to very poor | 17 (11.9) |
| Hyperglycemia (100mg/dL) | 55 (38.5) | | |
| Hypertriglyceridemia (150mg/dL) | 91 (63.6) | PHQ-9 score | 5.14±5.01 |
| High total cholesterol (200mg/dL) | 43 (30.1) | Severe depressive symptoms | 2 (1.4) |
| | | Moderate depressive symptoms | 25 (17.5) |
| Metabolic syndrome risk score | 2.70±1.34 | Mild depressive symptoms | 39 (27.3) |
| 0 | 8 (5.6) | No depressive symptoms | 77 (53.8) |
| 1 | 22 (15.4) | | |
| 2 | 30 (20.9) | Perceived social supports | 4.08±0.79 |
| 3 | 38 (26.5) | Emotional support | 4.20±0.79 |
| 4 | 34 (23.8) | Informational support | 3.96±0.99 |
| 5 | 11 (7.7) | Material support | 3.85±0.92 |
| | | Appraisal support | 4.21±0.75 |
| **Psychological characteristics** | | | |
| Subjective stress level | 2.85±0.88 | | |
| Very severe | 13 (9.1) | | |
| Severe | 29 (20.3) | | |
| Little | 68 (47.6) | | |
| None | 33 (23.0) | | |

Abbreviations: M = mean; MNT = Mongolian tugrik; SD = standard deviation

Abbreviations: M = mean; PHQ-9 = Patient Health Queationnaire-9; SD = standard deviation.

the PHW-9 criterion, 46.2% of the subjects had mild or more depressive symptoms, and 1.4% had severe depressive symptoms. The level of social support was generally high, with an average of 4.08±0.79 points out of 5, and the lowest subdomain being material support with 3.85 points.

## Comparison of socio-demographic, clinical and psychological characteristics according to metabolic syndrome risk score

The average and frequency analysis results for each characteristic variable of the subject according to the risk score for MetS are shown in Table 2. The characteristics of subjects with statistically significant differences between each score group for MetS were age ($p$ = .013), weight (< .001), BMI (< .001), abdominal circumference (< .001), and waist-hip ratio (WHR) (< .001), systolic/diastolic blood pressure (< .001), fasting blood sugar level (< .001), blood lipid index (< .001), overall body composition index (< .001 to .007), InBody score (.001), basal metabolic rate (.004), and stress perception level (.021). Among participants with a high-risk score for MetS, higher values were parallelly observed in their demographic and clinical characteristics. The main characteristics of the subjects when the MetS risk score was 3 or more were: age of 51 years or older, weight 70 kg, BMI of 28.8kg/m² or higher, abdominal circumference of 90 cm or higher, WHR of 0.97 or higher, systolic/diastolic blood pressure of 130/90mmHg or higher, fasting blood sugar level of 97mg/dL or higher, total cholesterol 176.5mg/dL or higher, triglyceride of 264.5mg/dL or higher, InBody score of 64.2 or less, basal metabolic rate 1,285kcal or higher, and over 3 points of stress perception were included. Among the psychosocial characteristics, there was no statistically significant difference since

**Table 2. Comparison of socio-demographic, clinical and psychological characteristics according to metabolic syndrome risk score (N = 143).**

| Variables | Category | Metabolic Syndrome Risk Score, n (%), M±SD | | | | | | $X^2$ / F | p |
|---|---|---|---|---|---|---|---|---|---|
| | | 0 (n = 8, 5.6%) | 1 (n = 22, 15.4%) | 2 (n = 30, 21.0%) | 3 (n = 38, 26.6%) | 4 (n = 34, 23.8%) | 5 (n = 11, 7.7%) | | |
| **Socio-demographics** | | | | | | | | | |
| Gender | Women | 8 (5.6) | 21 (14.7) | 23 (16.1) | 31 (21.7) | 28 (19.6) | 10 (7.0) | 5.634 | .344 |
| | Men | 0 (0.0) | 1 (0.7) | 7 (4.9) | 7 (4.9) | 6 (4.2) | 1 (0.7) | | |
| Age (years) | | 39.25±12.88 | 42.36±13.45 | 51.70±13.69 | 51.63±12.52 | 51.85±11.50 | 49.09±13.42 | 3.021 | .013 |
| Educational level | Middle school or less | 2 (1.4) | 3 (2.1) | 5 (3.5) | 6 (4.2) | 11 (7.7) | 2 (1.4) | 16.760 | 0.80 |
| | High school | 0 (0.0) | 9 (6.3) | 13 (9.1) | 19 (13.3) | 10 (7.0) | 1 (0.7) | | |
| | Diploma/Bachelors or above | 6 (4.2) | 10 (7.0) | 12 (8.4) | 13 (9.1) | 13 (9.1) | 8 (5.6) | | |
| Occupation | Yes | 8 (5.6) | 22 (15.4) | 27 (18.9) | 35 (24.5) | 32 (22.4) | 10 (7.0) | 3.035 | .695 |
| | No | 0 (0.0) | 0 (0.0) | 3 (2.1) | 3 (2.1) | 2 (1.4) | 1 (0.7) | | |
| Monthly incomes | (MNT[K]) | 881.2±435.8 | 472.7±271.6 | 594.1±432.3 | 646.5±487.3 | 626.4±431.3 | 375.1±337.0 | 1.856 | .106 |
| Marriage status | Living with the spouse | 8 (5.6) | 18 (12.6) | 24 (16.8) | 37 (25.9) | 30 (21.0) | 10 (7.0) | 7.295 | .200 |
| | Living without the spouse | 0 (0.0) | 4 (2.8) | 6 (4.2) | 1 (0.7) | 4 (2.8) | 1 (0.7) | | |
| Regular exercise | High intensity (yes) | 1 (0.7) | 4 (2.8) | 6 (4.2) | 7 (4.9) | 5 (3.5) | 0 (0.0) | 2.799 | .731 |
| | High intensity (no) | 7 (4.9) | 18 (12.6) | 24 (16.8) | 31 (21.7) | 29 (20.3) | 11 (7.7) | | |
| | Moderate intensity (yes) | 1 (0.7) | 11 (7.7) | 10 (7.0) | 13 (9.1) | 9 (6.3) | 1 (0.7) | 7.967 | .158 |
| | Moderate intensity (no) | 7 (4.9) | 11 (7.7) | 20 (14.0) | 25 (17.5) | 25 (17.5) | 10 (7.0) | | |
| Sleeping time (hours) | | 8.00±1.06 | 7.45±1.33 | 7.27±1.79 | 7.37±1.56 | 7.38±1.30 | 7.45±1.36 | .319 | .901 |
| **Clinical characteristics** | | | | | | | | | |
| Height (cm) | | 156.25±4.92 | 154.86±6.55 | 157.27±6.66 | 156.92±7.97 | 157.56±8.81 | 155.91±4.48 | .437 | .822 |
| Weight (kg) | | 55.87±4.61 | 56.77±9.53 | 68.00±13.26 | 70.84±11.96 | 76.67±15.79 | 73.00±17.45 | 8.077 | < .001 |
| Body mass index | | 22.95±1.60 | 23.70±3.19 | 27.40±4.45 | 28.81±4.05 | 30.76±4.90 | 30.57±5.49 | 10.668 | < .001 |
| (kg/m²) | <25.0 | 8 (5.6) | 15 (10.5) | 9 (6.3) | 6 (4.2) | 1 (0.7) | 0 (0.0) | 56.808 | < .001 |
| | ≥30.0 (overweight to obese) | 0 (0.0) | 7 (4.9) | 21 (14.7) | 32 (22.4) | 33 (23.1) | 11 (7.7) | | |
| Comorbidities | Total number | 0.38±0.74 | 0.91±1.06 | 1.00±1.17 | 1.13±1.14 | 1.00±0.81 | 1.00±0.77 | .749 | .588 |
| | ≥ 2 or more comorbidities | 1 (0.7) | 5 (3.5) | 7 (4.9) | 11 (7.7) | 9 (6.3) | 3 (2.1) | 1.149 | .950 |
| Current medications | Total number | 0.25±0.70 | 0.82±1.00 | 0.63±0.76 | 1.00±1.16 | 0.79±0.88 | 0.73±1.00 | 1.032 | .402 |
| | ≥ 2 or more medications | 1 (0.7) | 5 (3.5) | 3 (2.1) | 10 (7.0) | 8 (5.6) | 2 (1.4) | 3.450 | .631 |
| Waist circumference (cm) | | 77.37±2.77 | 77.81±9.48 | 88.13±10.88 | 90.71±11.11 | 93.88±10.37 | 94.18±10.49 | 9.441 | < .001 |
| Waist-hip ratio | Mean WHR | 0.90±0.03 | 0.90±0.04 | 0.94±0.07 | 0.97±0.07 | 0.98±0.06 | 0.99±0.05 | 6.727 | < .001 |
| | ≥0.90(men), 0.85 (women) | 8 (5.6) | 20 (14.0) | 26 (18.2) | 37 (25.9) | 34 (23.8) | 11 (7.7) | 59.024 | < .001 |
| | <0.90(men), 0.85 (women) | 0 (0.0) | 2 (1.4) | 4 (2.8) | 1 (0.7) | 0 (0.0) | 0 (0.0) | | |
| Systolic blood pressure | Mean SBP | 112.25±10.88 | 118.68±9.98 | 122.47±10.95 | 131.58±19.41 | 137.00±18.37 | 133.09±15.39 | 6.749 | < .001 |
| | ≥ 130mmHg | 0 (0.0) | 4 (2.8) | 9 (6.3) | 21 (14.7) | 17 (11.9) | 6 (4.2) | 17.041 | .004 |
| | < 130mmHg | 8 (5.6) | 18 (12.6) | 21 (14.7) | 17 (11.9) | 17 (11.9) | 5 (3.5) | | |
| Diastolic blood pressure | Mean DBP | 72.00±5.23 | 76.55±10.19 | 82.77±8.48 | 90.95±12.84 | 91.56±11.26 | 94.00±14.00 | 10.685 | < .001 |
| | ≥≥ 85mmHg | 0 (0.0) | 4 (2.8) | 12 (8.4) | 25 (17.5) | 27 (18.9) | 9 (6.3) | 37.494 | < .001 |
| | < 85mmHg | 8 (5.6) | 18 (12.6) | 18 (12.6) | 13 (9.1) | 7 (4.9) | 2 (1.4) | | |

(*Continued*)

**Table 2.** (Continued)

| Variables | Category | Metabolic Syndrome Risk Score, n (%), M±SD | | | | | | $X^2$ / F | $p$ |
|---|---|---|---|---|---|---|---|---|---|
| | | 0 (n = 8, 5.6%) | 1 (n = 22, 15.4%) | 2 (n = 30, 21.0%) | 3 (n = 38, 26.6%) | 4 (n = 34, 23.8%) | 5 (n = 11, 7.7%) | | |
| Fasting blood glucose | Mean FBS | 87.88±8.59 | 88.73±7.90 | 89.57±12.23 | 97.58±14.93 | 125.53±48.22 | 132.45±81.76 | 6.872 | < .001 |
| | $\geq$ 100mg/dL | 0 (0.0) | 0 (0.0) | 4 (2.8) | 14 (9.8) | 26 (18.2) | 11 (7.7) | 65.148 | < .001 |
| | < 100mg/dL | 8 (5.6) | 22 (15.4) | 26 (18.2) | 24 (16.8) | 8 (5.6) | 0 (0.0) | | |
| Total cholesterol | Mean Total cholesterol | 144.50±59.47 | 157.72±47.16 | 171.23±42.47 | 176.55±43.66 | 181.79±58.91 | 242.36±27.32 | 5.577 | < .001 |
| | $\geq$ 200mg/dL | 0 (0.0) | 1 (0.7) | 5 (3.5) | 12 (8.4) | 14 (9.8) | 11 (7.7) | 40.436 | < .001 |
| | < 200mg/dL | 8 (5.6) | 21 (14.7) | 25 (17.5) | 26 (18.2) | 20 (14.0) | 0 (0.0) | | |
| Triglyceride | Mean Triglyceride | 79.12±33.00 | 183.95±151.55 | 173.66±112.99 | 264.55±176.51 | 294.61±145.77 | 291.18±120.15 | 5.290 | < .001 |
| | $\geq$ 150mg/dL | 0 (0.0) | 8 (5.6) | 15 (10.5) | 25 (17.5) | 32 (22.4) | 11 (7.7) | 43.495 | < .001 |
| | < 150mg/dL | 8 (5.6) | 14 (9.8) | 15 (10.5) | 13 (9.1) | 2 (1.4) | 0 (0.0) | | |
| **Body composition** | Total body water (kg) | 26.92±2.12 | 27.39±3.89 | 31.09±5.22 | 31.18±5.14 | 32.86±6.67 | 31.20±5.67 | 3.720 | .003 |
| | Protein (kg) | 7.21±0.55 | 7.33±1.04 | 8.30±1.45 | 8.32±1.40 | 8.79±1.80 | 8.32±1.55 | 3.497 | < .001 |
| | Minerals (kg) | 2.52±0.24 | 2.54±0.38 | 2.90±0.44 | 2.89±0.47 | 3.01±0.60 | 2.88±0.45 | 3.361 | .007 |
| | Body fat mass (kg) | 19.40±3.13 | 19.80±5.87 | 25.77±9.09 | 28.70±8.59 | 31.92±9.94 | 32.31±10.78 | 7.631 | < .001 |
| | Free fat mass (kg) | 36.66±2.91 | 37.27±5.31 | 42.30±7.11 | 42.41±6.99 | 44.67±9.07 | 42.43±7.66 | 3.669 | .004 |
| | Skeletal muscle mass (kg) | 19.72±1.75 | 20.15±3.16 | 23.07±4.34 | 23.15±4.24 | 24.54±5.43 | 23.16±4.71 | 3.571 | .005 |
| InBody score | Total score | 69.62±3.29 | 69.81±4.73 | 66.46±7.10 | 64.21±6.58 | 63.02±6.87 | 62.36±6.96 | 4.609 | .001 |
| | < 70 (overweight/ sarcopenic) | 4 (2.8) | 9 (6.3) | 20 (14.0) | 28 (19.6) | 29 (20.3) | 9 (6.3) | 15.056 | .010 |
| | $\geq$ 70 (normal) | 4 (2.8) | 13 (9.1) | 10 (7.0) | 10 (7.0) | 5 (3.5) | 2 (1.4) | | |
| Body metabolic rate (kcal) | | 1162.00 ±63.23 | 1175.13 ±114.87 | 1283.60 ±153.83 | 1285.89 ±151.24 | 1335.02 ±196.08 | 1273.18 ±164.29 | 3.662 | .004 |
| **Psychological characteristics** | | | | | | | | | |
| Subjective stress level | | 2.50±1.06 | 2.36±0.72 | 2.73±0.90 | 3.03±0.75 | 3.03±0.96 | 3.18±0.75 | 2.746 | .021 |
| Self-rate health status | | 3.13±0.35 | 3.09±0.61 | 3.07±0.52 | 3.05±0.39 | 2.91±0.51 | 2.91±0.53 | 0.673 | .644 |
| PHQ-9 score | Total score | 7.12±4.12 | 6.72±4.87 | 5.70±5.40 | 4.60±5.36 | 4.35±4.74 | 3.36±3.66 | 1.311 | .263 |
| | Mild to Severe depression | 6 (4.2) | 14 (9.8) | 16 (11.2) | 13 (9.1) | 13 (9.1) | 4 (2.8) | 9.470 | .092 |
| Social supports | Total score | 3.98±0.83 | 4.11±0.67 | 4.16±0.81 | 4.08±0.79 | 3.92±0.92 | 4.34±0.51 | .598 | .701 |
| | Emotional support | 4.16±0.73 | 4.20±0.73 | 4.31±0.78 | 4.18±0.79 | 4.07±0.90 | 4.40±0.58 | .427 | .829 |
| | Informational support | 3.90±1.04 | 3.97±0.82 | 4.01±1.05 | 4.04±0.95 | 3.69±1.15 | 4.34±0.72 | .867 | .505 |
| | Material support | 3.71±1.00 | 3.93±0.78 | 3.98±0.97 | 3.82±0.93 | 3.65±1.02 | 4.18±0.54 | .778 | .567 |
| | Appraisal support | 4.05±0.72 | 4.24±0.79 | 4.26±0.70 | 4.20±0.76 | 4.14±0.88 | 4.41±0.45 | .598 | .701 |

Abbreviations: M = mean; MNT = Mongolian tugrik; SD = standard deviation.

the subjects were generally perceived to have a low level of depression and a high level of social support on average.

## Factor affecting metabolic syndrome risk score

In this study, it was found that all of the explanatory variables did not violate the assumption of parallel lines. In addition, after performing likelihood ratio verification to confirm the statistical significance of the model, the verification value was found to be 475.63 which was statistically significant at the significance level $p < .001$. The explanatory power of the model was 25.8% (Nagelkerke $R^2 = 0.249$) (Table 3). In the ordinal logistic regression analysis, age ($p$ =

**Table 3. Odds ratios of ordinal logistic regression model of metabolic syndrome risk score ($N = 143$).**

| Category | Variables | b | S.E | OR |
|---|---|---|---|---|
| Socio-demographics | Age | 1.4212** | .1528 | 2.021 |
| | Drinking status | .2479 | .1762 | 1.128 |
| | Smoking status | .0965 | .2814 | .9174 |
| | Regular exercise | -.6162* | .2523 | .5212 |
| Clinical characteristics | InBody score | -.5401** | .1820 | 1.872 |
| Psychological characteristics | Subjective stress level | .4270 | .2520 | 1.237 |
| | Self-rate health status | .5411 | .0172 | 1.310 |
| | Depression | .8570* | .0772 | 1.514 |
| | Social supports | -.2974 | .1624 | .9127 |
| Likelihood $x^2$ | | 475.63 *** | | |
| Nagelkerke $R^2$ | | 0.258 | | |

\* $p < .05$,

\*\* $p < .01$,

\*\*\* $p < .001$.

.002), regular exercise of moderate intensity or higher ($p = .032$), InBody score ($p = .001$), and depression level ($p = .022$) were statistically significantly associated with the risk score for MetS. The risk score for MetS increased by 2.02 times with age, and decreased by 0.52 times in those who exercised with more than moderate intensity. The risk score for MetS increased by 1.87 times as the InBody score decreased, and increased by 1.51 times in those with mild or more depressive symptoms compared to those without depression.

## Discussion

In this study, 58.0% of the subjects were found to have MetS. The previous epidemiological studies in various countries and on various races have reported an increasing trend in the prevalence of MetS [6, 27, 44]. The high risk of MetS in this study is thought to be due to the fact that subjects who already had comorbid diseases, such as cerebrovascular disease, cardiac disease, respiratory disease, hypertension, diabetes, and chronic liver disease, were included in the sample. In addition, the regional characteristics of living in rural areas in Mongolia (i.e., out of the city, with limited access to healthcare services), reflected in the results. Since this study was conducted in one area of inner Mongolia, there is a limit to directly comparing the prevalence of MetS in other countries or generalizing the health conditions of local residents.

Mongolia's healthcare system focuses more on hospitals and clinical treatment than on prevention and health promotion [36]. Furthermore, access to the health care system is low due to Mongolia's large territory and low population density [35, 36, 45]. Primary health care systems, such as family health centers, soum health centers, or intersoum hospitals, are partially operated at the local level, or bagh feldsher (i.e., where intermediate-level health personnel live in the ger; perform tasks such as home visit treatments, health promotion, and monitoring; and refer patients to soum hospital) [36, 45]. However, most of the secondary and tertiary medical institutions are concentrated in the capital city of Ulaanbaatar [36]. Private medical institutions are also mostly located in the capital. As medical facilities, human resources, and medical resources are concentrated in cities, adults living in rural areas have significantly less access to quality medical services, which is a major problem since they are often unable to effectively manage their health [35, 37].

Similar to previous studies [4, 9, 14, 17, 18, 31], the risk level of MetS was significantly different depending on the following variables: age, which is an uncorrectable variable; and factors that can be corrected, such as weight, BMI, abdominal circumference, blood pressure, blood sugar, dyslipidemia, body composition index, skeletal muscle mass, basal metabolic rate, and stress perception level. In addition, the factors affecting the risk score for MetS were age, regular exercise of moderate intensity or higher, InBody score reflecting obese or sarcopenic status, and depressive symptoms. This suggests that there is a need to focus on correctable factors in order to improve body composition and blood indicators, and to manage psychological health status. Since many previous studies [17, 30, 46] and clinical standard guidelines [2, 12, 40] emphasize that lifestyle improvement is more effective than medication therapy in preventing and managing MetS, it is necessary to provide interventions that focus on modifying lifestyle habits of adults in rural areas. It is necessary to lose weight and reduce body fat and abdominal fat through moderate or higher-intensity regular exercise, induce changes in body composition by introducing improved eating habits, and managing blood pressure, blood sugar, and dyslipidemia indicators.

According to the Mongolia Sustainable Development Vision 2030 report, four goals were set with the aim of ensuring healthy and quality living at all stages [35, 47]. Among these goals were measures such as reducing the disease burden of noncommunicable diseases, and reducing health risk factors and preventable deaths with participation and collaboration of families and local communities. The characteristics and infrastructure of local communities play an important mediating role in the health of residents. Thus, given that an individual's ability to manage their health and lifestyle is influenced by the community, several factors affecting health can be managed more effectively when approached from the local level. In this context, short-term strategies could consider resident participatory programs that engage and support rural community members in developing their human resources (e.g., health mentors). Among important long-term strategies, it is necessary to strengthen the competence of community health care professionals through systematic education and training for bagh feldsher and community nurses. Moreover, the status of rural primary health care services, such as family health centers, soum health centers, or intersoum hospitals, should be reviewed. In addition, it is necessary to establish a mid- to long-term health promotion policy by linking local networks with private resources, such as international volunteer organizations. As private projects, such as the official development assistant (ODA), often involve one-time visits or short-term programs, long-term and continuous intervention strategies must aim to support local residents' autonomy in health management. It is posited that such intervention strategies, consistently implemented over time, will be effective in reducing detrimental health and lifestyle practices.

In this study, the higher the risk score for MetS, the lower the InBody score—which was consistent with previous studies known to be associated with decreased muscle mass in cardiovascular diseases, such as coronary artery disease and diabetes [48, 49]. Although body weight does not change with age, a phenomenon wherein body fat increases but muscle mass decreases due to changes in body composition is known as sarcopenic obesity [50]. This decrease in muscle mass increases body fat by reducing physical activity and reducing calories consumed during rest, hindering the body's proper response to external stress [51, 52]. Thus, it is known to cause falls, dysfunction, increased hospitalization rates, decreased quality of life, and increased mortality [50].

Lim et al.'s research [51] on the association between MetS and the prevalence of sarcopenic obesity in a cohort study of elderly Koreans showed that the lower the muscle mass, the higher the risk of MetS. In this study, 35.1% of elderly men and 48.1% of elderly women had abdominal obesity and sarcopenic obesity. In particular, the risk of MetS was increased by 2.64 times

in the sarcopenia group, 5.51 times in the obese group, and 8.28 times in the case of both muscle loss and abdominal obesity [51]. These results suggest that practical strategies are needed to increase muscle mass and reduce abdominal obesity by encouraging moderate or higher-intensity regular exercise. In the present study, only 16.1% of respondents said that they would engage in high-intensity exercise regularly. Unlike in urban areas, farming and nomadic activities continue in rural areas of Mongolia because there are no major changes in life, such as retirement. In addition, compared to urban areas, infrastructures in rural areas, such as facilities and programs for health promotion, are relatively insufficient [35]. Rather than budget-intensive interventions, such as large-scale exercise facilities, equipment, and exercise experts, it is important to discover and utilize local resources (e.g., local autonomy committee, nearby school facilities, local residents) and develop and distribute home-based exercise intervention programs. It is also important to help rural residents recognize their own physical condition through regular body composition analysis. The body composition analysis used in this study used bioelectrical impedance, which has the advantages of no radiation exposure, a simple measurement method, and low cost. Thus, it is necessary to install such a body composition analyser in a local community health clinic or a family health center so that the body composition of local residents can be regularly analysed and managed.

It has been reported that stress increases insulin resistance [10], and depression increases abdominal obesity and worsens the inflammatory response—thereby increasing the incidence of MetS [53]. According to literature reviews [22, 53] that examined links between MetS and depression, most cross-sectional studies reported higher depression scores among subjects with MetS than those without the condition. It is consistent with the results of this study, which showed relatively higher depression scores in groups with greater risk for MetS. Therefore, it is imperative that the management of psychological concerns, such as stress and depression, are considered in interventions developed to support adults in rural areas.

Contrary to previous studies [20, 23, 54], however, the level of social support in this study did not affect the risk level of MetS. In previous studies, it was found that the sense of belonging as a community and a high level of social support promote physical and psychological health because they provides a sense of emotional stability in relationships among residents [23]. The finding that social support within the community affects the improvement of health outcomes (e.g., individual health level, well-being, and quality of life) are consistently supported by various studies [2, 19, 20, 23]. In particular, the higher the level of emotional and informational support, the higher the confidence and competence in personal health management [23, 42, 55]. The difference between the results of previous studies and this study is presumed to be due to the regional and cultural characteristics of Mongolia's rural areas, where the study was conducted. In the rural areas of Mongolia, adults continue to live close by their neighbors, have a nomadic lifestyle, and are subjected to harsh climate [37]. The social support perceived by local residents is relatively high. This high recognition of social support can act as a major disadvantage in situations where medical resources are insufficient; however, if sufficient levels of medical resources, manpower, and educational information are distributed, it is likely to serve as an advantage in improving the health level of residents. Lastly, most of the adults living in rural Mongolia had smartphones, but they did not obtain health information through their smartphones. Most of the educational materials, various documents and health-related materials provided by web services (e.g., YouTube) related to cardio-cerebrovascular health and MetS management are not produced in the Mongolian language. Therefore, it was impossible for rural residents of Mongolia to use such services. In order to improve their health level while overcoming the limitations of geographic access, it would be very useful to produce user-friendly educational materials in Mongolian and provide them to residents through information technology devices, such as smartphones.

This study has some limitations. First, as the convenient sampling method used in the study permitted participation of residents from specific rural areas, a limitation arises in generalizing and interpreting the research results for the total rural population of Mongolia. Certain measures were taken to encourage voluntary participation and address any study-related biases that participants may have. To minimize study bias that may occur due to voluntary participation, various methods were adopted. Local residents of the soum area (consisting of approximately 450 adults) were encouraged to participate through the cooperation of government officials who informed the residents about the study. In addition, participant convenience and comfort was considered and the main auditorium of the Office of Education, which has the best accessibility for local residents and is a largely familiar setting to them, was used for the purposes of data collection. Furthermore, following the advice of local Mongolians, data collection was conducted in July, which is regarded the best time for activities with minimal disturbances caused by extreme seasonal conditions, that also leads to movements characteristic of the local nomadic life. As the research team plans to conduct experimental interventions for the prevention and management of MetS for residents in this soum community in the future, efforts were made to collect basic data of as many residents as possible.

Second, due to the limitations of the cross-sectional descriptive research method, the longitudinal effects of factors contributing to the risk level of MetS could not be examined, and the health status of the subjects, such as the severity or duration of comorbidities, could not be controlled. Third, the total cholesterol level was used instead of the HDL level, which is the main diagnostic criterion for MetS. Due to the nature of the ODA project, there were environmental and administrative restrictions that made it difficult to bring all of the necessary diagnostic equipment and reagents into Mongolia. The overall explanatory power in this study was slightly low. In further research, a more accurate investigation of the various lifestyle habits that can affect MetS—such as eating habits, types of food intake, sleep patterns, and amount of exercise—should be undertaken. Despite these limitations, this study targeted adults living in rural Mongolian regions with insufficient medical infrastructure and low access to medical care. In addition, since biomarker and psychosocial indicators (especially those of social support, depression, etc.) are used together, the findings could contribute to constructing multi-faceted intervention programs for the management of MetS in the future. The results of this study are expected to be used as basic data for establishing strategies and policies for cost-effective health management of adults in rural areas of Mongolia.

## Conclusions

This study was conducted to explore the influence of factors such as socio-demographic characteristics, body composition, emotional state, and social support, on the risk of MetS in adults living in rural Mongolian regions. Age, regular exercise, InBody score, and depressive symptoms were found to influence the risk of MetS. Thus, in order to reduce the prevalence of MetS among adults in rural areas of Mongolia, it is important to help them improve their lifestyle by ensuring that they exercise regularly to reduce body fat and increase skeletal muscle. In addition, a strategy that provides both early evaluation of emotional states (e.g., depression), as well as interventions for effective management, will be helpful. It is very important to analyse the characteristics of rural areas of Mongolia with low access to health and medical resources, then establish and provide health promotion policies based on the needs and convenience of local residents.

## Supporting information

**S1 Appendix. The original questionnaire (Mongolian version).**
(PDF)

**S2 Appendix. The original questionnaire (Korean version).**
(PDF)

**S1 Dataset. Data of questionnaire.**
(XLS)

## Acknowledgments

The contributions of all participants in this study are greatly appreciated. We would like to thank Editage (www.editage.co.kr) for English language editing.

## Author Contributions

**Conceptualization:** Jin Hee Kim, Hyun Lye Kim, Jae Yong Yoo.

**Data curation:** Jin Hee Kim, Hyun Lye Kim, Bolorchimeg Battushig, Jae Yong Yoo.

**Formal analysis:** Jin Hee Kim, Hyun Lye Kim, Bolorchimeg Battushig, Jae Yong Yoo.

**Funding acquisition:** Jin Hee Kim.

**Investigation:** Jin Hee Kim, Hyun Lye Kim, Bolorchimeg Battushig, Jae Yong Yoo.

**Methodology:** Jin Hee Kim, Hyun Lye Kim, Bolorchimeg Battushig, Jae Yong Yoo.

**Project administration:** Jin Hee Kim.

**Resources:** Jin Hee Kim, Bolorchimeg Battushig.

**Software:** Jae Yong Yoo.

**Supervision:** Jin Hee Kim.

**Validation:** Jin Hee Kim, Hyun Lye Kim, Jae Yong Yoo.

**Writing – original draft:** Jin Hee Kim, Hyun Lye Kim, Jae Yong Yoo.

**Writing – review & editing:** Jin Hee Kim, Hyun Lye Kim, Jae Yong Yoo.

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
