## [Decision Letter · Decision Letter 0]

15 Mar 2021

PONE-D-21-00705

Effects of socio-demographics, body composition, emotional state, and social support on the risk level of metabolic syndrome of adults in rural Mongolia

PLOS ONE

Dear Dr. YOO,

Thank you for submitting your manuscript to PLOS ONE. After careful consideration, we feel that it has merit but does not fully meet PLOS ONE’s publication criteria as it currently stands. Therefore, we invite you to submit a revised version of the manuscript that addresses the points raised during the review process.

We look forward to receiving your revised manuscript.

Kind regards,

Frank T. Spradley

Academic Editor

PLOS ONE

3. Please include additional information regarding the survey or questionnaire used in the study and ensure that you have provided sufficient details that others could replicate the analyses. For instance, if you developed a questionnaire as part of this study and it is not under a copyright more restrictive than CC-BY, please include a copy, in both the original language and English, as Supporting Information. Moreover, please include more details on how the questionnaire was pre-tested, and whether it was validated.

Reviewers' comments:

Reviewer's Responses to Questions

**Comments to the Author**

1. Is the manuscript technically sound, and do the data support the conclusions?

Reviewer #1: Partly

2. Has the statistical analysis been performed appropriately and rigorously? 

Reviewer #1: Yes

3. Have the authors made all data underlying the findings in their manuscript fully available?

Reviewer #1: Yes

4. Is the manuscript presented in an intelligible fashion and written in standard English?

Reviewer #1: Yes

5. Review Comments to the Author

Reviewer #1: Kim et. al explored relationships between socio-demographics, clinical parameters, emotional state, and social support level on risk of prevalence of metabolic syndrome in Mongolia. Data was collected from voluntary clinic visits. The paper overall is well written; however there is concern with the stated goals and some of the conclusions. It is this reviewers suggestion to stick to the findings without making assumptions about how to reduce MetS in this area. In addition, the novelty was not clear as most of these correlations are well established.

Major Comments:

Line 92 (Stated goal of the paper): The study is all correlations in nature. Therefore, stating that the study investigated the effects of socio-demographics… on the risk of MetS prevalence can’t be the goal. This would imply there is an interventions.

Several of the conclusions are along the same lines. Without an intervention, you cannot say that a metastudy on depression and MetS that depression increases the prevalence of MetS without intervention. (Page 18, paragraph 3)

Conclusion paragraph: It is hard to say that outsiders should “…ensure that they exercise regularly to reduce body fat and increase skeletal muscle.” While these thing do improve pathophysiology related to obesity, this usually isn’t effective long term as people tend to go back to their old lifestyle at some point.

Minor Comments

In the first paragraph of the results, the significant correlations between metabolic risk score and other variables are stated, with a blanket statement for directionality (i.e. “variables tended to be significantly higher.” Authors should be definitive here.

Does the voluntary nature of the study bias the data given that the study subjects had to get to the clinic? It seems less mobile people might be excluded since they might be less likely to travel.

6. PLOS authors have the option to publish the peer review history of their article (what does this mean?). If published, this will include your full peer review and any attached files.

Reviewer #1: No

---

## [Author Response · Author response to Decision Letter 0]

15 May 2021

Authors' Response to academic editor's request

1) According to the editor's request, it was revised throughout the manuscript in compliance with PLOS ONE journal style standards.

2) We agree with PLOS ONE journal's data disclosure policy. According to the editor's opinion, we have attached data materials to the Supporting Information files.

3) We have attached the questionnaires (Mongolian version and Korean version) to the Supporting Information file at the request of the editor. The process of organizing the Mongolian version of the questionnaire was re-described in detail in the 'Measurements' section (Page 5, Line 119-126).

Authors' Response to reviewers' request

1) Thank you for your valuable opinion. We completely agree with you. Our research team focused on metabolic syndrome in Mongolian rural areas (soum in the aimag) among local residents with a special life culture that consists of small communities and combines nomadic life. In Mongolia’s health care system, medical facilities and medical staff are heavily concentrated in Ulaanbaatar, the capital of Mongolia. Therefore, this study was conducted as a preliminary research to plan practical strategies for improving the health level of rural residents in Mongolia. Most previous studies have been conducted in countries or regions with generally good access to medical facilities and medical staff, and studies in Mongolia have been mainly conducted in Ulaanbaatar. However, local residents who we actually met in Mongolia said they were new to body composition measuring equipment or said they had performed blood sugar checks or urine tests years ago despite having diabetes. Thus, by identifying the factors related to metabolic syndrome among local residents living in areas with insufficient medical infrastructure and low access to medical care, we considered the planning of a health management strategy with limited resources. This is the novelty of our study. In addition, since biomarker indicators and psychosocial indicators (especially social support, depression indicators, etc) are used together, we think they can contribute to the construction of a multi-faceted intervention program for the management of metabolic syndrome in the future. We will fully reflect the valuable opinions and suggestions of the reviewer when planning follow-up research. We fully agree with the reviewer’s comment. In the title, abstract, and purpose of the study, all expressions that can infer the “effects of an intervention” have been revised.

2) We fully agree with the reviewer’s comment. In the title, abstract, and purpose of the study, all expressions that can infer the “effects of an intervention” have been revised. In addition, the analysis results of previous review studies on the relationship between depression and metabolic syndrome have been described more clearly and amended. 

3) Thank you for your valuable opinion. 

We recognize that it is very important for members of the community to participate directly in order to manage the health level of local residents in the long term. Dahlgren & Whitehead’s “Rainbow model” of multi-layered influences of health also explains that factors of individual health behavior are not independent but are influenced by multi-layered influences (local community, socio-economic, cultural, and environmental conditions, etc.). In this sense, an individual’s health care ability is influenced by the community, and most of the factors affecting health are considered to be more effective when approached at the community level. We fully agree that there are limitations to improving and sustaining a healthy lifestyle among local residents with one-time/short-term projects, such as outsiders and the ODA program. In other words, it is necessary for local residents to experience the process of identifying and solving health problems on their own through “participation.” Through this, the health care capacity of local residents will increase, and it might become an independent activity with sustainability. According to the reviewer’s opinion, the content of the conclusion has been revised.

4) We re-described it specifically to improve the ambiguous representation of the research results. Methods to minimize study bias that may occur due to voluntary participation are described in detail in the limitations of the study.

Thank you for kind reviews.

---

## [Decision Letter · Decision Letter 1]

15 Jun 2021

PONE-D-21-00705R1

Relationship between socio-demographics, body composition, emotional state, and social support on metabolic syndrome risk among adults in rural Mongolia

PLOS ONE

Dear Dr. YOO,

Thank you for submitting your manuscript to PLOS ONE. After careful consideration, we feel that it has merit but does not fully meet PLOS ONE’s publication criteria as it currently stands. Therefore, we invite you to submit a revised version of the manuscript that addresses the points raised during the review process.

We look forward to receiving your revised manuscript.

Kind regards,

Frank T. Spradley

Academic Editor

PLOS ONE

Journal Requirements:

Reviewers' comments:

Reviewer's Responses to Questions

**Comments to the Author**

1. If the authors have adequately addressed your comments raised in a previous round of review and you feel that this manuscript is now acceptable for publication, you may indicate that here to bypass the “Comments to the Author” section, enter your conflict of interest statement in the “Confidential to Editor” section, and submit your "Accept" recommendation.

Reviewer #1: (No Response)

2. Is the manuscript technically sound, and do the data support the conclusions?

Reviewer #1: Yes

3. Has the statistical analysis been performed appropriately and rigorously? 

Reviewer #1: Yes

4. Have the authors made all data underlying the findings in their manuscript fully available?

Reviewer #1: Yes

5. Is the manuscript presented in an intelligible fashion and written in standard English?

Reviewer #1: Yes

6. Review Comments to the Author

Reviewer #1: The authors have addressed my comments mostly and improved the manuscript; however the conclusions in the abstract need to be about the data, not what the authors feel needs to happen to improve metabolic syndrome. Once this is addressed, I have no further comments and will recommend for publication.

7. PLOS authors have the option to publish the peer review history of their article (what does this mean?). If published, this will include your full peer review and any attached files.

Reviewer #1: No

---

## [Author Response · Author response to Decision Letter 1]

17 Jun 2021

Authors' response to academic editor's request: Thank you for your careful review.

We found that the one retracted article (Reference list: 25. Kaur JA. Comprehensive review on metabolic syndrome. Cardiol Res Pract. 2014;1-21. doi: 10.1155/2014/943162.) was included in a reference list. According to the editor's request, the retracted article removed and replaced with a new, relevant reference (Reference list: 25. Huang PL. A comprehensive definition for metabolic syndrome. Dis Model Mech. 2009;2(5-6):231-7. doi:10.1242/dmm.001180.). The revised reference list was marked in red.

Authors' response to reviewer's comment: Thank you for your valuable opinion. We fully agree with the reviewer’s comment. Based on the reviewer’s opinions, the conclusion in the abstract has been revised to focus on the main finding of this study. The revised sections were marked in red.

---

## [Editor Report · Decision Letter 2]

21 Jun 2021

Relationship between socio-demographics, body composition, emotional state, and social support on metabolic syndrome risk among adults in rural Mongolia

PONE-D-21-00705R2

Dear Dr. YOO,

We’re pleased to inform you that your manuscript has been judged scientifically suitable for publication and will be formally accepted for publication once it meets all outstanding technical requirements.

Kind regards,

Frank T. Spradley

Academic Editor

PLOS ONE

---

## [Editor Report · Acceptance letter]

23 Jun 2021

PONE-D-21-00705R2 

Relationship between socio-demographics, body composition, emotional state, and social support on metabolic syndrome risk among adults in rural Mongolia 

Dear Dr. Yoo:

I'm pleased to inform you that your manuscript has been deemed suitable for publication in PLOS ONE. Congratulations! Your manuscript is now with our production department. 

Kind regards, 

on behalf of

Dr. Frank T. Spradley 

Academic Editor

PLOS ONE